# Newborn Screening Protocols and Positive Predictive Value for Congenital Adrenal Hyperplasia Vary across the United States

**DOI:** 10.3390/ijns6020037

**Published:** 2020-05-08

**Authors:** Phyllis W. Speiser, Reeti Chawla, Ming Chen, Alicia Diaz-Thomas, Courtney Finlayson, Meilan M. Rutter, David E. Sandberg, Kim Shimy, Rashida Talib, Jane Cerise, Eric Vilain, Emmanuèle C. Délot

**Affiliations:** 1Division of Endocrinology, Cohen Children’s Medical Ctr of New York, Feinstein Institute for Medical Research, Zucker School of Medicine at Hofstra University, New Hyde Park, NY 11040, USA; rtalib@northwell.edu; 2Division of Endocrinology, Phoenix Children’s Hospital, Phoenix, AZ 85016, USA; rchawla@phoenixchildrens.com; 3Division of Endocrinology, CS Mott Children’s Hospital, University of Michigan, Ann Arbor, MI 48109, USA; chenming@med.umich.edu; 4Division of Endocrinology, LeBonheur Children’s Hospital, University of Tennessee Health Science Center, Memphis, TN 18103, USA; adiaztho@uthsc.edu; 5Division of Endocrinology, Ann & Robert H. Lurie Children’s Hospital of Chicago, Northwestern University Feinberg School of Medicine, Chicago, IL 60611, USA; cfinlayson@luriechildrens.org; 6Division of Endocrinology, Cincinnati Children’s Hospital Medical Center, University of Cincinnati, Cincinnati, OH 45229, USA; meilan.rutter@cchmc.org; 7Susan B. Meister Child Health Evaluation and Research Center, University of Michigan, Ann Arbor, MI 48109, USA; dsandber@med.umich.edu; 8Division of Endocrinology, Children’s National Medical Center, Washington, DC 20010, USA; kim.shimy@childrensnational.org; 9Feinstein Institute for Medical Research, Northwell Health, Manhasset, NY 11030, USA; jcerise@northwell.edu; 10Children’s National Hospital, Children’s Research Institute and George Washington University, Washington, DC 20010, USA; evilain@gwu.edu (E.V.); edelot@cnmc.org (E.C.D.)

**Keywords:** adrenal hyperplasia, congenital, newborn screening, standardization

## Abstract

Newborn screening for congenital adrenal hyperplasia (CAH) caused by 21-hydroxylase deficiency is mandated throughout the US. Filter paper blood specimens are assayed for 17-hydroxyprogesterone (17OHP). Prematurity, low birth weight, or critical illness cause falsely elevated results. The purpose of this report is to highlight differences in protocols among US state laboratories. We circulated a survey to state laboratory directors requesting qualitative and quantitative information about individual screening programs. Qualitative and quantitative information provided by 17 state programs were available for analysis. Disease prevalence ranged from 1:9941 to 1:28,661 live births. Four state laboratories mandated a second screen regardless of the initial screening results; most others did so for infants in intensive care units. All but one program utilized birthweight cut-points, but cutoffs varied widely: 17OHP values of 25 to 75 ng/mL for birthweights >2250–2500 g. The positive predictive values for normal birthweight infants varied from 0.7% to 50%, with the highest predictive values based in two of the states with a mandatory second screen. Data were unavailable for negative predictive values. These data imply differences in sensitivity and specificity in CAH screening in the US. Standardization of newborn screening protocols could improve the positive predictive value.

## 1. Introduction 

Newborn screening for congenital adrenal hyperplasia caused by steroid 21-hydroxylase deficiency (CAH-21) is mandated in the United States, 35 other countries, and in portions of 17 additional countries. Most of the US states report participation of 99.9% or higher [1]. The rationale for screening is to recognize and promptly treat the potentially life-threatening severe salt-wasting classic form of CAH. The overall prevalence of classic CAH-21 is approximately 1:14,000 to 1:18,000, of whom 75% have the salt-wasting form and 25% have non-salt-wasting or so-called simple virilizing CAH (summarized in [2]). Case fatality rates (lethality) of ~4% have been estimated for the era prior to newborn screening, presumably due to salt-wasting adrenal crisis, less frequent in the era of screening [3]. Affected 46,XY males cannot readily be identified without screening, as external genital virilization will flag the diagnosis only in 46,XX individuals. Lethality estimates were based on a rising disease prevalence, increased male to female case ratios, and increased salt wasting to simple virilizing case ratios observed since the advent of screening in the 1980s [4,5,6]. Other benefits of screening, aside from reduced infant mortality, have included less severe hyponatremia, shorter hospitalizations for infants identified with CAH [7,8,9,10], fewer learning disabilities as sequelae of salt-wasting crises [11,12], fewer reports of regret in the assigned gender of rearing [13,14], and potentially better growth outcomes [15]. Newborn screening protocols for measuring the typical marker steroid, 17-hydroxyprogesterone (17OHP), cut-points for abnormal values, and reporting of results are not standardized. Instead, each state develops and adapts its own procedures. The purpose of this report is to describe the varied CAH-21 screening practices in the US based on a sampling of individual state programs. 

## 2. Materials and Methods 

Pediatric endocrinologists and other members of the Disorders/Differences of Sex Development Translational Research Network (DSD-TRN) circulated a survey (Appendix A) to state laboratory directors of their own states and neighboring state programs requesting qualitative and quantitative information about CAH screening programs. Questions included basic epidemiologic statistics for the number of infants screened, number of positive screens, and number of cases confirmed as CAH. Details of the assay methods, cut-points, and factors considered for stratifying abnormal values were queried. We asked about the personnel responsible for reporting and receiving abnormal results for hospitalized and discharged infants, and criteria for follow-up testing. Data sets were returned from 17 states, and descriptive statistics were analyzed for the calendar year 2017. Results shown refer to those obtained in normal birthweight infants. Confirmed CAH cases (true positives) were defined as those reported to the state screening programs as meeting standard confirmatory diagnostic criteria for classic CAH, including a baseline serum 17OHP >10,000 ng/dL (or 300 nmol/L), cosyntropin testing resulting in markedly elevated stimulated serum 17OHP [16], and/or *CYP21A2* genotyping indicating known classic disease-causing mutations. False positives included those infants with 17OHP results above the cut-off point for birthweight but ultimately confirmed as not having CAH. Some of these data were presented in spring 2019 at The Endocrine Society Meeting, New Orleans, LA, USA and at the National Newborn Screening Board Meeting, Bethesda, MD, USA. 

### Statistical Methods

The reported number of screens, number of positive screens (excluding low birth weight infants), and number of endocrinologist-confirmed CAH cases from 2017 were used to estimate the number of true negative cases, prevalence, and positive predictive value (PPV). We arbitrarily assumed no false negative results for all programs, as negative screens were not referred for definitive diagnosis, and their final status was not reported to the state laboratories. Therefore, the number of confirmed cases was estimated for each state as the number of positive screens who were confirmed as having CAH, and the number of true negative cases was estimated for each state as the total number screened minus the sum of true positive and false positive cases. Under the assumption of no false negatives, the estimate of the number of confirmed cases served as a lower bound on the number of confirmed cases, and the estimate of the number of true negative cases served as an upper bound on the number of true negative cases. In other words, if there were any false negatives, this would result in a higher number of confirmed cases and a lower number of true negative cases. Similarly, the prevalence, estimated for each state as the number of confirmed positives divided by the total number screened, was a lower bound estimate. PPV for each state was estimated as 100 times the number of confirmed cases divided by the number of positive screens. The weighted mean prevalence was calculated as the sum of the prevalence for each state weighted by the proportion of patients screened for that state out of all 17 states. Under the assumption of no false negatives, because all newborns are screened, this is equal to the sum of all states’ confirmed cases divided by the sum of infants screened in all states.

Spearman correlation coefficients (shown as r_s_ below) were calculated between PPV and the 17OHP cut-off point, and between PPV and prevalence.

A *p*-value < 0.05 was considered statistically significant. All analyses were conducted using SAS version 9.4 (SAS Institute Inc., Cary, NC, USA). Figure 1 was created using the package ggplot2 in R version 3.6.1.

## 3. Results

### 3.1. Epidemiology

Completed questionnaires were provided by 17 states and were analyzed (Table 1). Partial responses from 3 additional states were not included. The single largest state screening population in our survey included >230,000 infants in a single year, and the smallest only 11,500. More cases were referred for follow-up and confirmatory testing, and then were verified as having CAH. States mandating a second screening sample referred only those second positive screens. In this survey, there were a total of 93 confirmed cases. The prevalence of CAH ranged from 1:9941 to 1:28,661 live births, with a weighted mean prevalence of 1:16,825, similar to published estimates (Table 1). We were unable to obtain mortality or morbidity statistics specific to CAH. 

### 3.2. Assays and Cut-Off Points

All laboratories used fluoroimmunoassay for 17OHP measurement in filter paper blood samples (DELFIA time-resolved fluorescence assay, Perkin Elmer, Waltham, MA, USA) obtained after 24 h of life. Most states performed the assay in their own laboratories, while some sent the samples to Perkin Elmer’s facilities using the same assay kits. All but one of the programs utilized birthweight cut-points, but cutoffs varied widely, from 17OHP values of 25 to 75 ng/mL (mean 41.2 ng/mL) for normal birthweights >2250–2500 g (Table 1). Cut-off points for lower birth weights were generally higher. Four state laboratories in our survey mandated a second screen regardless of initial screening results, and results of the second screen were used in this analysis where available. Most states utilized a later or second screen for infants in intensive care units, as early screening of sick, premature, or low birth weight infants gives many false positive results, contributing to lower positive predictive values [2].

The estimated positive predictive values for normal birthweight infants varied from 0.7% to 50% (mean 8.1%; median 5.3%). No statistically significant correlation was observed between PPV and the 17OHP cut-off point (r_s_ = −0.0059, *p* = 0.98, *n* = 16), nor between PPV and prevalence (r_s_ = 0.21, *p* = 0.42, *n* = 17). The highest predictive values were found in two of the four states with a mandatory second screen, with estimated PPV values greater than 1.5 times the interquartile range of 6.3. (Table 1, Figure 1).

### 3.3. Post-Analytical Procedures

There were no data concerning the cross-validation of assays carried out in different state laboratories. Each state reported positive results differently. Methods included phone calls, faxed documents, and/or mailed letters. Reporting went to a variety of reportees, including administrative aides, nurses, genetic counselors, or pediatric endocrinologists. Reports on infants with mildly or moderately abnormal screens who had been discharged from the birth hospital typically were sent to the primary care pediatrician listed on the hospital discharge. Information was not available regarding the content of counseling to the families. Infants with borderline or mildly elevated results usually underwent repeat screening. When an abnormal screen was obtained for a hospitalized infant, the neonatologist was informed, and a repeat sample obtained shortly thereafter. Those with more markedly abnormal results were most often referred directly to the pediatric endocrinologist at the state-designated center and underwent further confirmatory tests, such as a cosyntropin stimulation test if indicated, at the clinician’s discretion. We were unable to obtain data on the exact time between the reporting of results to the clinician and the start of treatment, as such data were not collected by the state laboratories. There was no consistent means of reporting infants missed by screening who later proved to be affected with classic CAH, and thus data were unavailable for false negative results or negative predictive values. Moreover, limited or no phenotypic data were recorded by state laboratories about the infants’ genital examination or other clinical features; no long-term outcomes were collected. Thus, we could not ascertain whether screening contributed to the prevention of salt wasting, nor were we able to distinguish salt-wasting from non-salt-wasting types of CAH.

## 4. Discussion

The primary aim of this report was to document differences in CAH newborn screening protocols. Our survey of about one-third of the US states confirms that screening for CAH using a single sample primary fluoroimmunoassay for 17OHP is associated with low positive predictive values [17]. In this report, we only included data related to the positive predictive value for normal birthweight infants, as prematurity, low birthweight, and sampling within the first 24 h of life are major causes of false positive results (reviewed in [2]). Although all laboratories used the same fluoroimmunoassay kit, conditions in the individual laboratories may have contributed to the lack of correlation between the cut-off point and positive predictive value. Cross-validation of assays might reduce some of the variability. Alternatively, these observations may be due to the relatively small number of confirmed positive cases in each state sample.

Protocols for CAH newborn screening have been adapted and implemented separately by each state laboratory in the US over the past several decades. Publications have generally reported data unique to an individual state screening program with a few exceptions [18,19]. For instance, at present, 13 states mandate a second screen at about two weeks of life in part due to the relatively low positive predictive value of the initial screen. These programs tend to show a higher disease prevalence compared with those only performing a single screen [18]. However, those later diagnosed infants are often non-salt-wasting, which some argue is not the objective of screening. In this survey, we could not ascertain whether screening contributed to the prevention of salt wasting, nor were we able to distinguish salt-wasting from non-salt-wasting types of CAH. According to the current Endocrine Society CAH Guidelines [2], all infants detected with CAH in the newborn period are to be treated with a combination of glucocorticoids, mineralocorticoid, and sodium chloride supplements. Thus, most pediatric endocrinologists no longer wait until evidence of salt wasting is observed before starting the patient on these standard treatments.

Alternative diagnostic methods and screening protocols have previously been evaluated. Suggestions for quality improvement include gestational age-based cut-off points [5,20] or a combination of birthweight and gestational age [21]. Implementation of a second screen at two weeks of life improves the positive predictive value as noted but adds cost and may be impractical or logistically difficult in some regions. A second tier test, specifically liquid chromatography-tandem mass spectrometry (LC-MS/MS), has been cited as a useful adjunctive test [22,23] that may be adapted to measure an extended steroid profile [24], although it is not currently suitable for high-volume screening at the first tier due to time constraints. At present, genotyping *CYP21A2* is not utilized in most US programs for either primary or secondary screening due to the complexity of test interpretation, time constraints, and costs [2], although this may change in the future.

Quality improvement efforts should be directed at enhanced communication between clinicians and state laboratories and transparent reporting of states’ efforts on a national platform. This would include not only the creation of common disease screening panels but also the standardization of protocols, with the aim of eventually achieving a uniform set of best practices [25,26]. With respect to the negative predictive value of screening, a centralized registry of CAH cases diagnosed after the newborn period would aid in capturing false negative tests from the newborn period. Colorado published data citing a 28% false negative rate in the primary screen once a second screen was implemented [27]. New York reported only three false negative cases from their single screen over a span of 7 years, but this is highly dependent on complete follow-up and physician reporting [19]. Surprisingly, Minnesota found a 32% false negative rate, even in the context of two screens [28]. The causes of such disparities are unclear and should be investigated. Finally, there are negative psychological implications to conveying inaccurate screening results to parents, and improving these protocols could have important mental health ramifications. Nationwide (NewSteps [29] and the Disorders/Differences of Sex Development-Translational Research Network (DSD-TRN). [30,31]) and international (Clinical and Laboratory Standards Institute [32]) collaboratives offer opportunities to explore and compare screening outcomes and develop best practices.

## Figures and Tables

**Figure 1 IJNS-06-00037-f001:**
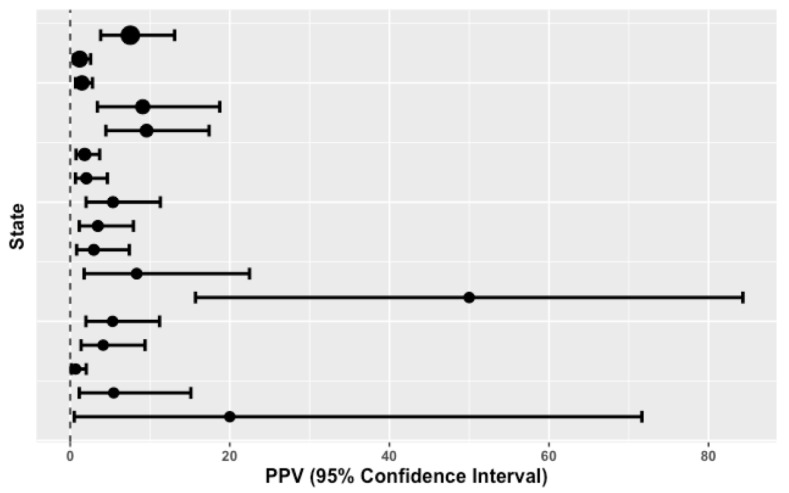
The positive predictive values (PPVs) calculated for each state are shown, with their respective 95% confidence intervals. States are ordered from the smallest to largest number of screens performed (from bottom up on *y* axis). Dot size is related to the number of screens performed (numerical data used are shown in Table 1).

**Table 1 IJNS-06-00037-t001:** CAH newborn screening by state.

#Screened (Total = 1,564,756)	#Referred Positive (Total = 3217)	#Confirmed CAH (Total = 93)	Prevalence (Weighted Mean = 1:16,825)	PPV (95% CI) IQR = [2.02 to 8.33]	Birth Weight Cut-Off Point (Grams)	17OHP Cut-Off Point (ng/mL) (Mean = 41.2)	Two-Screens Mandate
230,431	146	11	1:20,948	7.53 (3.82–13.08)	>2251	35	N
171,964	506	6	1:28,661	1.19 (0.44–2.56)	>2500	55	N
138,226	608	9	1:15,358	1.48 (0.68–2.79)	>2500	70	N
135,590	66	6	1:22,598	9.09 (3.41–18.74)	>2500	30	N
109,740	94	9	1:12,193	9.57 (4.47–17.4)	>2500	65	N
104,000	387	7	1:14,857	1.81 (0.73–3.69)	>2250	25	N
84,000	247	5	1:16,800	2.02 (0.66–4.66)	>2500	30	N
81,117	112	6	1:13,520	5.36 (1.99–11.30)	>2500	50	Y
79,948	144	5	1:15,990	3.47 (1.14–7.92)	>2249	37	N
79,000	135	4	1:19,750	2.96 (0.81–7.41)	≥2500	35	Y
72,440	36	3	1:24,147	8.33 (1.75–22.47)	>2251	75	N
61,500	8	4	1:15,375	50.00 (15.70–84.30)	>2500	25	Y
59,643	113	6	1:9941	5.31 (1.97–11.2)	No values given		N
55,935	121	5	1:11,187	4.13 (1.36–9.38)	>2500	25	N
53,361	434	3	1:17,787	0.69 (0.14–2.01)	>2500	30	N
36,361	55	3	1:12,120	5.45 (1.14–15.12)	≥2500	38.3	N
11,500	5	1	1:11,500	20.00 (0.51–71.64)	≥2300	35	Y

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
