# Peer review of "Newborn Screening Protocols and Positive Predictive Value for Congenital Adrenal Hyperplasia Vary across the United States"

_2409-515X, 2020, doi:10.3390/ijns6020037_

Round 1

Reviewer 1 Report

This is an interisting paper aiming to evaluate the different newborn screening protocols in the US. For this purpose the authors used a simple online questionnaire to evaluate sreeningsmethods and follow up. The authors describe impressive diferences between US stated in terms of cut off levels and PPV and they conslude that screening has to be improved by standardization of protocols.

Unfortunately, the authors describedonly some  of the results in the text. For the reader it would be interisting to get more information about the results of 6 points in the questionnaire.

The authors found surprisingly big differences in PPV between the states but it is still unclear whether this can be explained by methodological or other factors. Probably, in some countries there are a lot of CAH patients that were missed  or not sreened and present with salt wasting or precocious puberty. Mortality rate is not reported. What is the participating rate in each state? So what is the birth rate in each state?

Furthermore, it would be interisting to know whether the screening contributed to help  prevention of salt wasting. So what is the time period from screening until  treatment. A screening does not make sense when the results or diagnoses is still delayed.

Do the authors have information about SV CAH children?

Reviewer 2 Report

The authors  carried out a statistical analysis of data collected in many newborn screening laboratory of US for CAH, highlighting many differences in protocols and outcomes. Presented data demonstrated that there are differences in sensitivity and specificity in CAH screening, and that the test positive predictive value should be improved with the standardization of newborn screening protocols. I find the work well written and organized and the questionnaire is also really interesting. Only a few errors require the reviewer of the manuscript.

Line 57: “CAH (summarized in [1].”

Line 124:  the sentence “States mandating a second screening sample referred only those second positive screens" is repeated.

Line 148: “Each state reported positive results differently: Method…”

Round 2

Reviewer 1 Report

The authors responded adequatley to most of the suggestions.

Althought the intention of the authors is to evaluate the outcomes of neonatal screening in fact only the screenings procedure  is evaluated. As there is no information about the clinical data such as preventing salt wasting ( This is THE main indicaton for screening) morbidity and mortality I would suggest to adapt the title and to delete the term "outcomes:  as this term is misleading.

Author Response

May 5, 2020

Manuscript ID: IJNS-775481 – Revision 2

Speiser PW, et al. Newborn screening protocols and outcomes for congenital adrenal hyperplasia vary across the United States

We herewith respectfully submit our point-by-point responses to reviewers (italicized text has been added to the appropriate section of the ms):

Responses to Reviewer 1, Round 2:

Reviewer 1, Round 2, Point 1: Althought the intention of the authors is to evaluate the outcomes of neonatal screening in fact only the screenings procedure  is evaluated. As there is no information about the clinical data such as preventing salt wasting ( This is THE main indicaton for screening) morbidity and mortality I would suggest to adapt the title and to delete the term "outcomes:  as this term is misleading.

Response: We have changed the ms title according to the reviewer’s suggestion to: “Newborn screening protocols and positive predictive value for congenital adrenal hyperplasia vary across the United States.” Further, we have omitted any mention of our findings as “outcomes.” This word may be taken to mean “clinical outcomes,” however, in our report the meaning is “performance characteristics.” We hope this will clarify any uncertainty. We agree with this reviewer that a registry including long-term follow up of CAH cases would be ideal, however, this is not currently feasible in the US.

No other changes were made to the text of the ms, figure or table. Please see attached revised version of our ms below. Thank you for your helpful suggestions.

Sincerely,

Phyllis W. Speiser, MD

Email: pspeiser@northwell.edu